# The Use of Oncolytic Viruses in the Treatment of Multiple Myeloma

**DOI:** 10.3390/cancers13225687

**Published:** 2021-11-13

**Authors:** Georgia Stewart, Andrew Chantry, Michelle Lawson

**Affiliations:** 1Sheffield Myeloma Research Team, Department of Oncology and Metabolism, University of Sheffield Medical School, Sheffield S10 2RX, UK; grstewart1@sheffield.ac.uk (G.S.); a.d.chantry@sheffield.ac.uk (A.C.); 2Mellanby Centre for Musculoskeletal Research, University of Sheffield Medical School, Sheffield S10 2RX, UK; 3Department of Haematology, Sheffield Teaching Hospitals NHS Foundation Trust, Royal Hallamshire Hospital, Sheffield S10 21F, UK

**Keywords:** multiple myeloma, oncolytic viruses, DNA viruses, RNA viruses, reovirus, MV-NIS and VSV-IFNβ-NIS

## Abstract

**Simple Summary:**

Multiple myeloma is a type of blood cancer caused by the uncontrolled growth of antibody producing B cells (known as plasma cells) that reside in the bone marrow. It is classed as a largely incurable cancer as whilst patients respond well to initial chemotherapy treatments, unfortunately after periods of disease remission, relapse usually occurs with the emergence of chemotherapy resistance. Therefore, there is a need for new approaches that not only reduce tumour load but also prevent tumour relapse. Oncolytic viruses (OVs) (tumour killing viruses) are being explored as a therapy for various cancers, including multiple myeloma. This review discusses the use of OVs in myeloma in preclinical model systems and early phase clinical trials, and discusses some of the hurdles involved in the translation to myeloma patients.

**Abstract:**

Multiple myeloma accounts for 1% of all new cancers worldwide. It is the second most common haematological malignancy and has a low five-year survival rate (53.2%). Myeloma remains an incurable disease and is caused by the growth of malignant plasma cells in the bone marrow. Current anti-myeloma therapies (conventional chemotherapies, immunomodulatory drugs i.e., thalidomide and its’ analogues, proteasome inhibitors, monoclonal antibodies, and radiotherapy) initially substantially debulk tumour burden, but after a period of remission ‘plateau phase’ disease invariably relapses due to tumour recrudescence from foci of minimal residual disease (MRD) and accumulating drug resistance. Therefore, there is a compelling clinical need for the development of novel treatment regimens to target MRD and effectively eliminate all remaining tumour cells. This review will discuss the potential use of oncolytic virus (OV) therapies in the treatment of myeloma. Specifically, it will focus on preclinical studies using DNA viruses (adenovirus (Ad), vaccinia virus (VV), myxoma virus (MYXV), and herpes simplex virus (HSV)), RNA viruses (reovirus (reo), coxsackie virus, measles virus (MV) and bovine viral diarrhoea virus (BVDV), and vesicular stomatitis virus (VSV)), and on four types of viruses (VV, reo, MV-NIS and VSV-IFNβ-NIS) that have been assessed clinically in a small number of myeloma patients.

## 1. Introduction

Multiple myeloma accounts for 1% of all new cancers worldwide and it is the second most common haematological malignancy [1]. A neoplasm of differentiated B cells, myeloma is characterised by the hyperproliferation of malignant plasma cells (MPCs) primarily in the bone marrow (>10%). Clinical manifestations include skeletal events, anaemia, and immunodeficiency [2]. In addition, most myeloma patients develop renal impairment, due to the accumulation of immunoglobulin free light chains (FLC) secreted from the MPCs.

Over the last 20 years there have been significant changes in the landscape of anti-myeloma therapies due to a better understanding of its pathobiology. This has led to the introduction of more effective therapies which include immunomodulatory drugs (IMiDs), e.g., lenalidomide [3]; proteasome inhibitors (PIs), e.g., bortezomib [4]; and more recently, histone deacetylase inhibitors (HDACi), e.g., panobinostat [5], monoclonal antibodies (mAbs), e.g., daratumumab [6], isatuximab (anti-CD138) [7], and elotuzumab (anti CS1/SLAMF7) [8], resulting in improvements in progression-free and overall survival rates. CAR-T cell therapy, including B cell mature antigen (BCMA)-targeted CAR-Ts [9] and bi-specific T cell engagers (BiTES) antibodies [10], is promising, with emerging evidence of efficacy. However, overall survival rates remain relatively short, with ~52.3% of myeloma patients surviving only five years [1]. In addition, myeloma patients still typically experience reduced treatment-free intervals between sequential therapies. Treatment refractory disease remains common and eventually death with resistant disease is almost inevitable [11]. Therefore, effective therapies that provide either a durable disease control or a complete eradication are urgently needed. This review will discuss the potential use of oncolytic virus (OV) therapies in the treatment of myeloma. Specifically, it will focus on preclinical studies using DNA viruses (adenovirus (Ad), vaccinia virus (VV), myxoma virus (MYXV) and herpes simplex virus (HSV)), RNA viruses (reovirus (reo), coxsackie virus, measles virus (MV), bovine viral diarrhoea virus (BVDV), and vesicular stomatitis virus (VSV)), and on four types of viruses (VV, reo, MV-NIS and VSV-IFNβ-NIS) that have been assessed clinically in a small numbers of myeloma patients.

## 2. Oncolytic Virotherapy in Cancer

Novel therapies that do not rely on the use of generic chemotherapies or targeted small molecule therapies either with their inevitable toxicities and/or drug resistance are highly desirable for the treatment of myeloma. This is because two of the major factors contributing to therapeutic failure in myeloma are tumour heterogeneity and plasticity leading to the emergence of resistant clones [11]. Oncolytic virotherapy is a promising therapeutic strategy with the potential to eradicate cancer cells with live, replicating viruses. Interest in this area first started when anecdotal evidence emerged in the 20th century that viral infection could induce remission in various cancer types [12,13], including myeloma [14]. Since these first observations, several viruses have been investigated and developed as therapeutic agents for cancer. The first OV, a second-generation herpes simplex virus type 1 (HSV-1) armed with granulocyte macrophage-colony stimulating factor (GM-CSF), obtained US FDA and the European Medicine Agency approvals in 2015 for melanoma [15]. Since then, a paradigm shift has occurred where some OVs are now being used in patients for some cancers, and it is likely that they will be accepted into mainstream clinical practice when used in combination with other anti-cancer therapies.

OVs can infect both normal and malignant cells, but malignant cells provide a superior environment due to the presence of aberrant signalling pathways, abnormal homeostasis, and responses to stress, which are advantageous for viral replication, such as overexpressed surface attachment receptors, activated RAS or Akt, or defective IFN pathways. [16,17]. These aberrant signalling pathways are activated during malignant transformation and antagonise host antiviral responses, creating cancer cells that are hypersensitive to viral infection and replication, which actively kills the cancer cells by lysis. One example is Reolysin, a wildtype (WT) reo that preferentially replicates in RAS-transformed cells [18,19,20]. This type of cell death is widely accepted to be highly immunogenic as dying cancer cells create a pro-inflammatory environment which causes activation of systemic innate and tumour-specific adaptive immune responses (Figure 1) [21]. Thus, OVs can be considered a double-armed therapeutic strategy where cancer cells are killed by direct viral oncolysis or are targeted by the immune system. This makes OVs superior to conventional therapies due to the detrimental effects chemotherapy has on the immune system. Both cell death mechanisms are beneficial. However, they are also linked to some therapeutic hurdles. For example, direct oncolysis can be rapid and efficient in some settings, but elimination occurs only in cells directly infected with a virus. Therefore, delivering the virus to sites of tumour is of huge importance, and this is a challenge that remains unsolved in the field. On the other hand, activation of anti-tumor immunity can entirely sterilize the site of tumour, but it is much slower than direct oncolysis, and frequently only shows clinical benefit in a small percentage of patients [22] including those with myeloma [14,23]. Therefore, successful oncolytic virotherapy requires a fine balance between the two mechanisms of targeting tumour cells. However, OVs can be armed with additional transgenes that either enhance cancer cell death or anti-tumour responses.

## 3. The Use of Oncolytic Viruses in the Treatment of Multiple Myeloma

OVs have been exploited for their anti-tumor effects across a range of solid and haematological malignancies. Specifically, in myeloma, a single case report from 1987 suggested that VV could (at least transiently) reduce systemic tumour burden in a patient [14]. Since then, preclinical studies have focused on a range of DNA and RNA viruses which have shown efficacy in myeloma, with viral infection resulting in the rapid elimination of MPCs through a variety of mechanisms (Table 1). Thus far, preclinical success with VV, reo, MV, and VSV has led to clinical studies in a small number of myeloma patients (Table 2).

### 3.1. The Use of DNA Oncolytic Viruses in the Treatment of Myeloma

#### 3.1.1. Preclinical Studies with Adenovirus in the Treatment of Myeloma

Ads are non-enveloped dsDNA viruses that have been used widely for gene therapy. An early study demonstrated that transduction with Ad vectors, using tumour selective DF3 promoter to drive the expression of thymidine kinase (TK), efficiently transduced myeloma cell lines [24]. Fernandes et al. [25] examined myeloma targeted delivery of CD40L by a conditionally replicative OVAd (AdEHCD40L) and demonstrated that myeloma cells were susceptible to AdEHCD40L-mediated apoptosis, and subsequently showed in a xenograft mouse model that AdEHCD40L treatment reduced tumour burden by 50%.

Given that most haematological malignancies do not express high levels of coxsackievirus and adenovirus receptor (CAR), the main entry receptor for Ad5, it has not generally been considered as an oncolytic agent for malignancies such as myeloma. However, Senac et al. [26] showed that Ad5 can infect and kill most myeloma cell lines and ex vivo patient MPCs, suggesting it has therapeutic potential. Meanwhile, Drouin et al. [27] looked at Ad5 against a chimeric Ad5/F35 in B lymphocytes. The latter substitutes Ad5-based vector fibres with those of serotype Ad35, which uses CD46 as a receptor. This study found that adenoviral transduction efficiencies varied between B-cell lines corresponding to distinct stages of B-cell lymphogenesis. Cell lines of plasma origin were efficiently transduced with both Ads, and interestingly, plasma cell lines with high transduction efficacy had the lowest CAR expression, whilst cells with high CAR expression had low Ad5 transduction efficiency. Moreover, CD46 was shown to be ubiquitously expressed, yet some cells remained poorly permissive to Ad5/F35. Collectively, these results showed that there was no direct correlation between transduction efficiency and cellular receptor expression for both Ad5 viruses. A further study showed that αvβ_5_ integrin is the primary receptor for Ads in CAR-negative cells [28] and this could explain the high transduction efficiencies in low CAR expressing cells.

Early work showed that WT Ad5 infects myeloma cell lines, but the life cycle is delayed compared to that of permissive cells [26,29] potentially due to repression of the E1 transcripts in myeloma cells [29,30]. This suggests that modifying E1 transcripts to evade destabilisation will enhance the efficacy of Ads in myeloma [26]. More recently, a study investigated the use of Lokon oncolytic Ad (LOAd) therapy for myeloma treatment [31]. They used Ad5/35 to target the virus to infect CD46 positive cells, then 2 LOAds, LOAd700 and LOAd703, were further modified to express immunomodulatory transgenes, where both LOAds encoded for CD40L, but LOAd703 also encoded for 4-1BBL. A panel of myeloma cell lines were sensitive to both LOAds, resulting in replication and cell death in vitro. When myeloma cells were co-cultured with healthy donor peripheral blood mononuclear cell (PBMCs) and treated with LOAds, LOAds promoted activation of cytotoxic T cells, and IFNγ expression. However, it would be interesting to see if these increases also occur in myeloma patient PBMCs or bone marrow mononuclear cells, who are typically immunosuppressed. In an in vivo study using a subcutaneous xenograft myeloma model, only mice given LOAds via intratumoral injection had a significant reduction in tumour burden compared to control, and LOAd703 treated mice had significantly longer overall survival [31].

Most in vivo studies using oncolytic Ads have been performed in immunodeficient mice. To fully test the potential therapeutic activity of OVs, studies should be performed in immunocompetent animals that can support active viral infection [32]. However, the stringent species selectivity of adenoviridae permits human Ad to infect rodent cells but does not permit replication. Despite this, some groups have found mouse cell lines that are permissive of Ad infection and replication [32,33,34].

Despite the preclinical activity of Ads in myeloma cell lines and mouse models, clinical investigation has not yet been realised, and the efficacy and safety profile need to be defined in phase I/II clinical trials.

#### 3.1.2. Preclinical Studies with Vaccinia Virus in the Treatment of Myeloma

VV is a large dsDNA virus that belongs to the Poxviridae family, derived from the original cowpox or horsepox virus with the historical use as a smallpox vaccine with proven clinical safety in humans. However, to better target myeloma cells, VVs have been genetically modified. TK and vaccinia growth factor (VGF) genes were double-deleted in a highly attenuated VV (VVDD) [35]. VVDD preferentially replicated in dividing cells and infected a range of myeloma cell lines, leading to decreased cell viability compared to controls in vitro. In myeloma xenograft models, treated intravenously with VVDD, significant tumour reduction and improved survival were observed compared to controls. More recently, a TK-deleted VV strain was engineered to express one of two anti-tumour factors, miR-34a (VV-miR-34a) and Sma (VV-Sma), with the former inhibiting several oncogenic processes and the latter involved in apoptosis induction [36]. Both viruses showed increased efficacy in vitro and in vivo compared to parental VV, but the combination of both viruses showed the most efficacy, where synergy and induction of apoptosis through the caspase pathway was observed. The same authors also modified a TK-deleted VV to express beclin-1 (VV-BECN-1), an essential autophagy protein linked to multiple processes including tumour suppression [37]. VV-BECN1 showed efficacy in vitro and in vivo using a myeloma cell line but did not cause cytotoxicity in PBMCs. VV-BECN-1 induced autophagy, but not apoptosis, in myeloma cells, through activation of sirtuin1 (*SIRT1*).

#### 3.1.3. Clinical Use of Vaccinia Virus in the Treatment of Myeloma

In the late 1980s, a single case study of a 67-year-old Japanese male patient with IgA myeloma was reported. Intravenous injection of VV (AS strain) markedly reduced IgA levels from 1309 mg/dL to 432 mg/dL after 96 days of treatment with no adverse effects reported [14]. However, a full phase I clinical trial should be conducted before drawing conclusions about VV’s safety and efficacy in myeloma patients.

#### 3.1.4. Preclinical Studies with Myxoma Virus in the Treatment of Myeloma

MYXV is a non-segmented dsDNA with a strict tropism for rabbits and hares. Whilst MYXV does not infect healthy non-malignant human cells, MYXV has been shown to infect a variety of cancer cell types. In myeloma cells, MYXV induced significant rapid oncolysis that was dependent upon caspase-8 mediated apoptosis and inhibited *ATF4* expression during the unfolded protein response [38,39]. In a xenograft model of myeloma, intravenous injection of MYXV resulted in rapid debulking of tumour (70–90%), but it is important to note that these effects may not be as pronounced in a more advanced stage of disease. Additionally, MYVX induced an anti-myeloma CD8^+^ T cell response which resulted in a significant overall survival. Due to MYXV rapid induction of oncolysis, it has been proposed that MYXV may be an effective purging strategy for autologous stem cell transplants (ASCTs) [40]. Arming murine allogeneic bone marrow containing a mouse myeloma cell line with MYXV and transplanting into recipient mice dramatically ablated pre-seeded residual myeloma in vivo. Additionally, MYXV was able to eliminate CD138^+^ myeloma cells from patient bone marrow samples, but whether the same effects are seen across a wider range of heterogeneous myeloma patients need to be explored. More recently, autologous murine bone marrow carrier leukocytes, pre-armed with MYXV, were therapeutically superior to MYXV armed PBMCs or free virus [41]. Additionally, when survivor mice were rechallenged with the same myeloma, they developed immunity. Currently, due to limited data, there have been no clinical trials with MYVX in myeloma patients.

#### 3.1.5. Preclinical Studies with Herpes Simplex Virus in the Treatment of Myeloma

HSV is dsDNA virus belonging to Herpesviridae family and although HSV has shown success in melanoma [15], it has only recently been investigated in myeloma [42]. In vitro HSV-1 infected myeloma cell lines and CD138^+^ primary cells and caused cell death independent of HSV-1 replication, due to apoptosis. In a subcutaneous xenograft myeloma model, HSV-1 treatment decreased tumour volume after intratumoral injection [42]. Another study reported similar findings in vitro and examined if the presence of an immune cell population would enhance the cytotoxic effect of HSV-1 [43]. Co-cultures of myeloma cell lines and PBMC cells (from healthy donors or myeloma patients) were treated with HSV-1. The addition of PBMCs significantly increased the cytotoxicity of HSV-1 and increased IFN-α and IFN-β secretion from PBMCs. Blocking IFN-α with antibodies or depleting plasmacytoid dendritic cells (DCs) or natural killer (NK) cells decreased the enhanced HSV-1 induced cell death in the presence of PBMCs. When HSV-1 was combined with lenalidomide, enhanced anti-myeloma effects were observed. However, when HSV-1 was combined with lenalidomide and IFN-α, this combination resulted in the greatest enhancement of cell death (43). These two recent studies provide evidence that HSV-1 may have clinical potential in myeloma.

### 3.2. The Use of RNA Oncolytic Viruses in the Treatment of Myeloma

#### 3.2.1. Preclinical Studies with Reovirus in the Treatment of Myeloma

Reo is a dsRNA virus of the Reoviridae family of viruses, with the human type 3 Dearing strain being developed for oncolytic virotherapy. Reo has shown efficacy against myeloma cell lines and ex vivo tumour samples [44,45], where cell death occurs through apoptosis [45], although upregulation of autophagy genes is also observed [44]. In vivo, reo established no effect on human CD34^+^ stem cells, and demonstrated complete eradication of myeloma cells, preventing relapse and improved survival in mice [45]. Reolysin (a proprietary formulation of WT reo) combined with bortezomib decreased tumour burden and bone disease in xenografts models of myeloma with no adverse effects observed [46]. However, reo sensitivity has been shown to vary in myeloma cell lines, some cells are highly sensitive (RMPI-8226, U226), while others are less sensitive (H929, L-363, MM.1S) [47]. Interestingly, reo sensitive myeloma cells have higher JAM-A expression with higher levels of reo genome observed compared to less sensitive myeloma cells. JAM-A is epigenetically regulated in cells, so it can be modulated by HDACi. Myeloma cells treated with a HDACi increased JAM-A expression by epigenetically regulating its promoter through increased histone acetylation and RNA polymerase II recruitment. Reolysin combined with HDACi enhanced myeloma oncolysis in vitro and in vivo, presenting a potential tool to increase reo efficacy [47].

More recently, reo has been shown to increase PD-L1 expression in MPC lines in vitro and in vivo, this increase was not observed when UV-inactivated reo was used [48]. When reo was given in conjunction with anti-PD-L1 therapy in the syngeneic 5TGM1 murine model of myeloma, combination therapy enhanced anti-myeloma efficacy by decreasing tumour burden and enhancing survival compared to either therapy alone. Therefore, reo and PD-1/PD-L1 targeted therapy could be beneficial for myeloma patients. In vitro, reo has been combined with bortezomib where synergistic interactions were observed in bortezomib-resistant cell lines [49]. In vivo, in a syngeneic Vk*MYC bortezomib-resistant murine myeloma model, enhanced anti-myeloma activity such as decreased tumour burden and improved overall survival was observed when reo was used in combination with bortezomib. Mechanistically, bortezomib augmented reo replication in myeloma cells and tumour-associated endothelial cells, as assessed by increased reo protein levels. The study also showed enhanced anti-myeloma immune responses following combination treatment such as increased CD3^+^ T cells, NK cells, PD-L1 expression and decreased T-regs and tumour associated macrophages (TAMS) [49]. Reo has also been shown to reduce tumour burden and bone disease in a mouse model of myeloma, augmenting anti-myeloma immune responses [50]. In the 5TGM1 syngeneic myeloma model, reo treatment increased NK cell and CD8^+^ T cell numbers and activation, and upregulated effector-memory CD8^+^ T cells. Additionally, the study found that co-culture of myeloma cells with bone marrow stromal cells (BMSCs) was able to induce resistance to MPC reo oncolysis and bystander cytokine killing, but the BMSCs were not able to protect the MPCs from reo-activated NK cells and myeloma-specific cytotoxic T cells [50].

#### 3.2.2. Clinical Use of Reovirus in the Treatment of Myeloma

Twelve patients with symptomatic relapsed and refractory myeloma, with previous bortezomib and lenalidomide treatment with or without previous ASCT, were administered Reoylsin intravenously in a dose-escalating, single centre phase I clinical trial [23]. Reolysin was well tolerated, no dose-limiting toxicities were experienced, but some grade three toxicities were noted. Reoviral RNA and protein was found in bone marrow biopsies. The longest duration of stable disease was eight months, but there was no significant disease response. This could be a result of low JAM-A expression, viral resistance, inadequate dosing, or limited viral-mediated anti-tumor immune response [51]. Currently, a phase Ib study is recruiting for reo in combination with bortezomib and dexamethasone in patients with relapsed or refractory myeloma (University of Southern California, NCT02514382).

#### 3.2.3. Preclinical Studies with Coxsackie Virus in the Treatment of Myeloma

Coxsackie virus is a non-enveloped positive-sense ssRNA virus, with the most studied virus being CVA21, but limited work has been done in myeloma. Au et al. [52] showed myeloma cell lines express of both ICAM-1 and DAF, with a strong association between ICAM-1 and CD138^+^ cells in myeloma patient bone marrow biopsies, whilst PBMCs expressed low levels of ICAM-1. CVA21 caused cytopathic effects in myeloma cells, but these were minimal in PBMCs compared to untreated controls, thought to be due to differences in viral replication. CVA21 purging of myeloma bone marrow biopsies varied, but bone marrow progenitor cells were not significantly affected [52]. Currently, there have been no clinical trials with coxsackie in myeloma patients due to lack of data.

#### 3.2.4. Preclinical Studies with Measles Virus in the Treatment of Myeloma

MV is an enveloped negative-sense ssRNA virus in the family Paramyxoviridae. The most studied strain is the attenuated Edmonston strain (MV-Edm), which has mutations in two accessory proteins (C and V) leading to tumour selectivity, genetic stability and is non-transmissible [53]. In vitro, a GFP-tagged MV-Edm was able to effectively lyse myeloma cells, whilst having no effect on phytohemagglutinin (PHA)-stimulated peripheral blood lymphocytes [54]. In vivo, intratumoral MV-Edm treatment of subcutaneous ARH-77 tumours resulted in all tumours regressing. Intravenous administration had a similar effect [54]. MV-Edm was further modified to encode human thyroidal sodium/iodine symporter (MV-NIS) and infection of myeloma cells with MV-NIS showed oncolysis in vitro [55]. In vivo, three subcutaneous xenograft models were tested (two MV-sensitive, one MV-resistant). The two MV-sensitive xenografts regressed completely after one intravenous dose of MV-NIS. The MV-resistant xenograft was unresponsive to MV-NIS infection alone, but when combined with iodine-123 (123I), resulted in enhanced tumour regression compared to MV-NIS [55]. These studies have limitations, as xenograft models typically have a defective immune response, they were conducted in a subcutaneous setting, and it is possible that 123I could suppress normal haematopoiesis in patients. Therefore, to better target MV to myeloma, the H protein of an MV variant was mutated so it lacked the ability to bind to its receptors CD46 and SLAM. Hummel et al. [56] then attached a single chain variable fragment (scFV) based on a mouse mAb known as Wue-1 which binds to CD138, which is expressed on healthy plasma cells and primary MPCs. MV-Wue infected and killed primary MPCs but also infected healthy plasma cells which could be a concern. However, in lymphoma patients, the widespread use of the well tolerated anti-CD20 mAb rituximab causes long lasting B cell depletion after administration. Therefore, eradicating plasma cells after MV-Wue administration should not generate toxicity problems.

#### 3.2.5. Clinical Use of Measles Virus in the Treatment of Myeloma

A phase I clinical trial with systemically delivered MV (MV-NIS) in patients with recurrent or refractory myeloma was published in 2017 [57]. The trial identified some grade III and IV haematological toxicities. Despite this, however, some interesting results were obtained, with one patient undergoing complete disease regression, whilst other patients had variable and transient drops in their serum FLCs. However, the existence of anti-MV antibodies in most patients who have been vaccinated against the virus potentially negates its oncolytic potential and limits its clinical use. In agreement with this, further investigation was done post-trial which found that the patient who achieved complete remission following MV-NIS had a low baseline titre of anti-MV antibodies, high baseline counts of both MV-reactive and TAA-reactive T cells and a high mutational burden. This patient subsequently had two focal relapses, at nine- and 30-months post MV therapy, which were successfully treated with radiotherapy. Therefore, the authors speculate that the long-term remission observed in this patient is because of sustained immune control of residual myeloma, driven by their high mutational burden, causing more expression of TAAs that were targeted by cytotoxic T cells. A phase II clinical trial is being conducted with MV-NIS in combination with cyclophosphamide in myeloma patients (NCT02192775). The trail involves administering a single intravenous dose of MV-NIS followed by a four-day course of cyclophosphamide. The trial has been completed but data have not been published yet.

#### 3.2.6. Preclinical Studies with Bovine Viral Diarrhea Virus in the Treatment of Myeloma

BVDV is a ss-RNA virus belonging to the Flaviviridae family, and is a major viral pathogen of cattle [58]. Myeloma cell lines have been shown to express CD46, the BVDV receptor, treatment of these cells with BVDV resulted in infection and cell death by apoptosis [59]. More importantly, BVDV decreased CD138^+^ primary myeloma cells from patient bone marrow aspirates and did not affect CD3^+^, CD19^+^, or CD56^+^ cell populations, suggesting that the BVDV oncolytic effect was limited to CD138^+^ myeloma cells. In vivo, in a subcutaneous xenograft myeloma mouse model, mice were treated with BVDV intratumorally, which led to a reduction in tumour volume, and an increase in caspase-3 activity. Additionally, the authors showed pretreatment with bortezomib increased BVDV efficacy in myeloma cell lines in vitro and suggested this was due to the activation of caspase-3-meidated apoptosis [59]. Due to these limited data, there have been no clinical trials with BVDV in myeloma patients.

#### 3.2.7. Preclinical Studies with Vesicular Stomatitis Virus

VSV is a member of the family Rhabdoviridae and is an enveloped negative-sense ssRNA virus that commonly infects livestock animals, but it can infect humans causing flu-like illnesses. Preclinically, VSV has demonstrated effectiveness against myeloma in vitro and in vivo [60]. VSV was engineered to express NIS, allowing for treatment with radioactive iodine and to track virus via imaging [61]. VSV-NIS was able to replicate to high titres in MPC lines and cause oncolysis in MPC lines and primary MM cells in vitro. In vivo, in a subcutaneous xenograft model, VSV-NIS showed high intratumoral viral replication which resulted in tumour regression. In a 5TGM1 syngeneic murine model with either subcutaneous or orthotopic tumours, enhanced tumour regression and survival were achieved when VSV-NIS was combined with radioactive iodine [61]. To enhance VSV oncolysis and improve safety, VSV was engineered to express IFN-β. VSV-IFNβ-NIS significantly improved anti-myeloma responses and prolonged survival compared to treatment with control VSV in subcutaneous and disseminated 5TGM1 syngeneic models of myeloma [62].

#### 3.2.8. Clinical Use of VSV-IFNβ-NIS in the Treatment of Myeloma

Preclinical results with VSV have led to the establishment of an early phase clinical trial (NCT03017820) with VSV-IFNβ-NIS in patients with haematological malignancies, including relapsed myeloma. The trial is currently active and recruiting.

## 4. Discussion

Multiple myeloma poses considerable therapeutic challenges to be overcome if oncolytic virotherapy is to translate to an effective treatment strategy. The field suffers from several obvious issues, which must be addressed. Firstly, the impact of disease heterogeneity needs to be examined in the context of oncolytic virotherapy. A review of the literature showed that preclinical results derive more from tests on myeloma cell lines than on patient-derived primary myeloma cells [17]. However, studies have shown that OVs in primary myeloma patient samples can infect and typically kill a high percentage of cells. Unfortunately, these studies used a small numbers of patient samples (typically only data from one to three patients) [22], with the exception of one more recent paper (*n* = 31) [59]. The field would therefore benefit from a comprehensive study of the efficacy of OVs in larger cohorts of myeloma patient samples.

Secondly, most of the literature studying OVs in myeloma administer OVs intratumorally in immunodeficient models of a single subcutaneous plasmacytoma. Although these studies represent an easy starting point, they are not clinically relevant myeloma models, and fail to recapitulate the clinical realities in two ways. First, due to the nature of OV, there are different treatment challenges associated with systemic versus localised disease. In particular, delivery of a virus through the circulation to sites of systemic disease is a major translational hurdle [22]. Since myeloma is typically a systemic disease, this should be addressed in preclinical models. Secondly, the immune system is recognised to have significant positive and negative impacts on OVs. For example, complement erythrocyte ‘virus traps’ and antiviral antibodies can inhibit viral infectivity [22,63,64,65]. On the other hand, much of OV efficacy is thought to be mediated through the induction of anti-tumor T-cell-mediated immunotherapy [22,66]. However, myeloma patients have a high degree of immune dysregulation, either due to the immunosuppressive effects of the disease or chemotherapy [67,68]. Myeloma and its immunosuppressive environment, makes it problematic to achieve sufficient immune responses with OVs alone [67]. Therefore, OVs should be studied in immunocompetent systemic disease models, however this poses an issue with some OVs, which cannot replicate in murine cells, such as the Ads and MV [69]. It must be noted that the lack of immunocompetent mouse models has not stopped the MV progressing into clinical trials in myeloma. Nonetheless, even recently, subcutaneous solitary plasmacytoma models are still used over systemic disease models for oncolytic virotherapy in myeloma [31,59].

Lastly, combining OVs with more typical myeloma standards of care therapies, such as chemotherapy, radiation or mAbs is lacking but may be the best approach to increase efficacy. Results from the first completed clinical trial utilising oncolytic reo in myeloma support this notion [23]. Enhanced anti-tumour immune responses have been demonstrated preclinically and clinically in other cancers when combining chemotherapy and oncolytic virotherapy [70,71,72,73,74,75,76]. This is critical in determining oncolytic virotherapy in context of other myeloma therapies. For example, should OVs be used as a frontline therapy? Or are they better suited to relapsed/refractory patients? Is there any synergism with OVs and existing myeloma treatments? As discussed earlier, one of the main issues with patients relapsing is the existence of MRD, so can OVs remove MRD once a tumour has been debulked by existing myeloma chemotherapies? This is a novel and more clinically relevant strategy to target these remaining cells that persist in MRD. Therefore, if the field is to advance from preclinical work to successful clinical trials, this will largely depend on the ability to use synergistic treatment approaches along with administering OVs at the most clinically relevant time points to eliminate MRD, which, if successful, could be curative for myeloma patients.

## 5. Conclusions

Multiple myeloma remains a predominantly incurable disease, despite current therapies debulking 90% or more of the initial tumour burden. Patients usually succumb to cycles of disease relapse due to disease recrudescence from foci of MRD and the emergence of chemo-refractory clones of MPCs [11]. Therefore, there is an urgent clinical need for the development of novel treatment approaches to eliminate all tumour cells. Here, we have discussed the use of OVs in the treatment of multiple myeloma, including several preclinical studies as well as clinical studies using reo, MV-NIS and VSV-IFNB-NIS. Although many unknowns and barriers remain in the field of onco-virotherapy, over the coming years the use of OVs in myeloma has the potential to lead to prolonged remission periods or even total eradication of MRD, i.e., curative therapy.

## Figures and Tables

**Figure 1 cancers-13-05687-f001:**
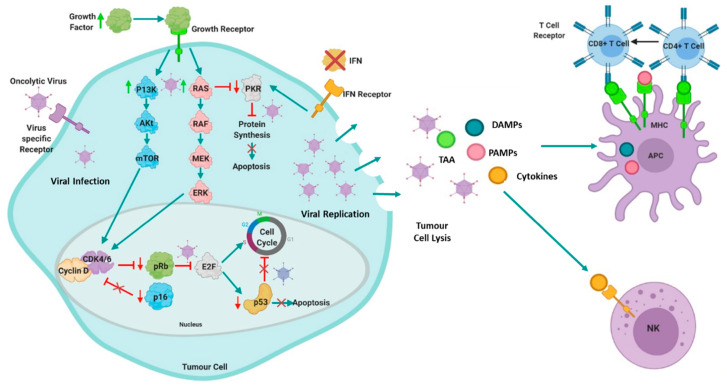
Mechanisms of action of oncolytic viruses: Cancer cells provide a superior environment for oncolytic viral replication due to the presence of aberrant signalling pathways or targeted molecules (such as RAS, PKR, P13K/Akt/mTOR, IFN, p53 or Rb/E2F/p16). Hyperactivation of P13K/Akt/mTOR and RAS pathways and consequent activation of their downstream cascade results in an increase of cell proliferation and inhibition of apoptosis. Oncolytic viruses can exploit this to replicate efficiently. Additionally, hyperactivation of RAS blocks protein kinase R (PKR), which prevents cells from detecting stress, thus allowing the cells to continue protein translation and abberrant proliferation, consequently, oncolytic viruses can selectively replicate as cancer cells cannot terminate protein translation in response to infection. Interferon (IFN) signalling may also be abnormal in malignant cells, thus viral clearance is prevented. Malignant cells with mutated p53 cannot undergo abortive apoptosis, therefore some viruses preferentially target p53 mutant cells. Likewise, aberrant expression of cell cycle regulatory components such as retinoblastoma protein (pRb) and p16, can render cancer cells sensitive to oncolytic viruses. Furthermore, oncolytic viruses have a natural tropism for receptors expressed abnormally in cancer cells [17]. These abberrant cancer pathways and/or targeted molecules allow for the OV to replicate relatively unchallenged resulting in cell death via passive oncolysis or other active cell death mechanisms. Following oncolysis, cytokines, tumour-associated antigens (TAAs), and other danger signals, such as damage associated molecular patterns (DAMPS) molecules and pathogen-associated molecular pattern (PAMPs) molecules are released which activates the systemic innate and tumour-specific adaptive immune response. The release of TAAs, in combination with the local cytokines and DAMPS, activates natural killer (NK) cells and stimulates antigen presenting cell (APC) maturation, which in turn, triggers both CD4^+^ and CD8^+^ T Cells in the adaptive immune response by cross-presentation [17]. Figure created using BioRender.

**Table 1 cancers-13-05687-t001:** Preclinical development of oncolytic viruses for multiple myeloma.

Oncolytic Virus	Genome	Enveloped	Replication Site	Receptors for MM	Mechanisms of Specificity	Mechanisms of Killing	Genetic Manipulation	Combination Therapy
DNA Viruses								
Adenovirus	dsDNA	No	Nuc/Cyto	UnK/CAR	UnK	Lytic Viral Replication	Easy	N/A
Vaccinia	dsDNA	Yes	Cyto	UnK	Engineered	Lytic viral replication	Easy	N/A
Myxoma	dsDNA	Yes	Cyto	UnK	Myeloma specific binding (UK receptor)	Induction of apoptosis	Easy	N/A
HSV	dsDNA	Yes	Nuc/Cyto	HVEM	UnK	Induction of apoptosis	Easy	Len
RNA Viruses								
Reo	dsRNA	No	Cyto	JAM-A	OE of JAM-A	Lytic viral replication, apoptosis, autophagyUPR	Difficult	BTZ, Len, Pom, anti-PD-L1
Coxsackie	ss(+)RNA	No	Cyto	ICAM-1, DAF	Unk	Unk	Easy	N/A
MV	ss(−)RNA	Yes	Cyto	CD46	OE of CD46	Lytic viral replication	Easy	CP
BVDV	ss(+)RNA	Yes	Cyto	CD46	OE of CD46	Induction of apoptosis	Unk	BTZ
VSV	ss(−)RNA	Yes	Cyto	LDLRs	Defects in interferon response	Lytic viral replication Inhibition of DNA synthesis	Easy	BTZ, CP
NDV	ss(−)RNA	Yes	Cyto	Sialic acids	UnK	Lytic viral replication, induction of apoptosis	Unk	N/A
Rotavirus	dsRNA	No	Cyto	PDI, HSP, integrin β3	UnK	Lytic viral replication	UnK	N/A

Abbreviations: MM: Multiple myeloma; HSV: Herpes simplex virus; MV: Measles virus; VSV: Vesicular stomatitis virus; BVDV: Bovine viral diarrhoea virus; NDV: Newcastle disease virus; Cyto: Cytoplasm; Nuc: Nucleus; OE: Overexpression; UPR: Unfolded protein response; UnK: Unknown; N/A: not applicable; BTZ: Bortezomib; Len: Lenalidomide; CP: cyclophosphamide. Positive-sense (+) or negative-sense (−), single-stranded (ss) RNA virus or double-stranded (ds) DNA virus.

**Table 2 cancers-13-05687-t002:** Clinical trials of oncolytic virotherapy in multiple myeloma.

Virotherapy	Clinical Trial Phase	Combination Agents	Clinicaltrials.Gov Identifier
Reovirus (Reolysin)	I	None	NCT01533194 (Completed)
	I	Lenalidomide or Pomalidomide	NCT03015922 (Active)
	I	Bortezomib & Dexamethasone	NCT02514382 (Unknown)
	I	Carfilzomib & Dexamethasone	NCT02101944 (Recruiting)
Measles virus (MV-NIS)	I/II	± Cyclophosphamide	NCT00450814 (Completed)
	II	Cyclophosphamide	NCT02192775 (Completed)
	I	± Cyclophosphamide	NCT00450814 (Completed)
VSV (VSV-IFNB-NIS)	I	± Cyclophosphamide	NCT03017820 (Recruiting)

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
