# Peer review of "The Use of Oncolytic Viruses in the Treatment of Multiple Myeloma"

_cancers, 2021, doi:10.3390/cancers13225687_

Round 1
Reviewer 1 Report
Multiple myeloma accounts for 1% of all new cancers worldwide, it is the second most 11 common haematological malignancy and has a low 5-year survival rate (53.2%). Myeloma remains 12 an incurable disease and is caused by the growth of malignant plasma cells in the bone marrow. There is a compelling clinical need for the development of 18 novel treatment regimens to target MRD and effectively eliminate all remaining tumor cells.
The authors need to better state how oncolytic therapies are superior to the currently available treatments for recurrent MM.
What are the drawbacks and limitations of oncolytic therapies for MM.
There are many typos throughout the text.
Author Response
We thank the reviewer for their critical assessment of the review.
We have expanded on why OVs are potentially superior to conventional treatments (lines 138-150).
We have expanded the review with a “Discussion” section to include the limitations of OV therapies in the treatment of multiple myeloma (lines 495-557).
We apologise for the typos, we have now checked the review article thoroughly and amended accordingly.
Reviewer 2 Report
The manuscript submitted for review is a review of the literature on the possible treatment of multiple myeloma with oncolytic viruses. The review is an interesting collection of reports on the above topic and is a very interesting aspect of the future of myeloma therapy. The manuscript is presented in a neat form. It seems, however, that it has already been explored by other authors. In March 2021, the publication of Marchica et al. or, quoted by the authors in item 45, the publication of Louise M E Müller, also from 2021, or many others. Recently, many publications on the above topic have been published in journals with a much lower impact index. It, therefore, seems inappropriate to publish a low innovation review manuscript in a journal with as much impact as Cancers.
Author Response
We thank the reviewer for their critical assessment of the review. Although other articles have been recently written in this subject area, this was in invited review in a special issue titled “Emerging Treatment Strategies for Multiple Myeloma” and we believe it is important to include this topic in this special issue.
We have expanded the review with a “Discussion” section to include the limitations of OV therapies in the treatment of multiple myeloma (lines 495-557).
Reviewer 3 Report
The authors have provided a comprehensive overview of the various oncolytic virus approaches that have been applied to the treatment of multiple myeloma. It is largely well written and logically structured. The article lacks a critical discussion or academic proposals for what should be the way forward. As such the article is a summative account of the history of the field but does not assist or provoke debate. It has done little to persuade this reader that this is a field on the cusp of success.
I do not follow the statement that OV’s show tropism toward malignant cells due to the presence of aberrant signalling pathways, abnormal homeostasis and response to stress which are advantageous to for viral replication. This may be because I misunderstand the term tropism in a viral context. I understand that once infected cancer cells may present a superior environment for viral replication than their normal counterpart, but isn’t that a slightly different concept to tropism?
Similarly, in the legend to figure 1 ‘oncolytic viruses preferentially select cancer cells’ suggests some act of will by the virus and needs re-wording
I would have liked some more description of the biology to understand how viruses ‘exploit’ the cancer cell environment.
More detail in table 2 would be helpful to inform the reader if all the trials are completed, whether some are ongoing and whether completed trials have been published.
Lines 258-9: font issue
Line 276: can the authors expand on the reasons why there have been no clinical trials?
Lin 354: typo: limitations, as
Line 368: ‘despite this some impressive results’; The word impressive appears out of context with the rest of the article where such terms have not been used. Impressive compared to what?
Author Response
We thank the reviewer for their critical assessment of the review.
We agree with your points regarding tropism and adding further detail on how viruses preferentially select cancer cells. Therefore, we have now changed the word tropism and given an example of how viruses exploit the cancer cell environment (lines 128-131 and this is also included in the Figure 1 legend). We have only given one example as different viruses use different mechanisms and not all mechanisms are completely known.
We have updated Table 2 to show which trials have been “Completed” or are “Recruiting” or “Active”.
We have corrected/amended the text with regards to font issues and typos. We have changed the word impressive to “interesting”.
The reason some viruses have not yet progressed into myeloma clinical trials is due to limited preclinical data or the data has only been published recently and not yet had enough time to progress into a trial. We have amended the text to reflex these reasons.